# Effect of Different Welding Methods on Flip-Chip LED (FC-LED) Filament Properties

**Mengtian Li [1], Jun Zou [2,3,4,*], Wengjuan Wu [2,*], Mingming Shi [2], Bobo Yang [2,5], Wenbo Li [3] and Bin Guo [6]**

[1] School of Material Science and Engineering, Shanghai Institute of Technology, Shanghai 201418, China; 166081125@mail.sit.edu.cn

[2] School of Science, Shanghai Institute of Technology, Shanghai 201418, China; mmshi@sit.edu.cn (M.S.); boboyang@sit.edu.cn (B.Y.)

[3] Zhejiang Emitting Optoelectronic Technology Co., Ltd., Jiaxing 314100, China; liwenbo@emitting.cn

[4] Institute of New Materials & Industrial Technology, WenZhou University, Wenzhou 325000, China

[5] Institute of Beyond Lighting, Academy for Engineering and Technology, Fudan University, Shanghai 200433, China

[6] School of Electronics and Information Engineering, Changchun University of Science and Technology, Changchun 130022, China; guobin@cust.edu.cn

* Correspondence: zoujun@sit.edu.cn (J.Z.); wwj@sit.edu.cn (W.W.)

**Abstract:** This paper investigates the effect of two different welding methods, direct welding (DW) and vacuum furnace welding (VFW), on flip-chip light-emitting diode (FC-LED) filament properties. Shearing force, SEM, steady-state voltage, steady-state luminous flux, and change of photoelectric performance with aging time were employed to characterize the differences in filament properties between the two welding methods. The shearing test revealed that the average shearing force of the VFW group was higher than that of the DW group, but the two groups followed the standard. Furthermore, the microstructure of the VFW group fault was more smoother, and the voids were fewer and smaller based on the SEM test results. The steady-state voltage and luminous flux revealed that the VFW group had a more concentrated voltage and a higher luminous flux. The aging data revealed that the steady-state voltage change rate of both groups was not very different, and both luminous flux maintenance rates of the VFW group were higher than those of the DW group, but all were within the standard range. In conclusion, if there is a higher requirement for filament in a practical application, such as the filament is connected in series or in parallel and needs a higher luminous flux, it can be welded using vacuum furnace welding. If the focus is on production efficiency and the high performance of filaments is not required, direct welding can be used.

**Keywords:** flip-chip LED; welding methods; microstructure; aging; photoelectric performance

## 1. Introduction

The light-emitting diode (LED), which is the most promising cold light source in the 21st century due to its energy-saving, environmental protection, high reliability, and flexible design qualities and other advantages, has been widely developed and applied in the field of lighting [1]. Furthermore, the flip-chip LED (FC-LED) filament, which is a new generation of LED lighting sources because of its flat coating technology, 360° luminescence, no blue light, long life, and slow decay, is a popular choice among those who recall older forms of lighting. Additionally, in terms of appearance, FC-LED filament bulbs are closer to incandescent lamps. With the state ban on incandescent lamps, FC-LED filament bulbs will gradually replace incandescent lamps [2–4]. In contrast with the ordinary filament, the FC-LED filament, which has a flexible aluminum substrate, can be bent at any angle to satisfy the

requirements of different shapes. During transport, the FC-LED filament will not break easily and will not stop functioning when used, which greatly reduces transportation costs and improves the service life of the filament [5,6].

Although the welding line is omitted in the process description on FC-LED filament packaging, solid crystal welding is still an extremely important part of the process. Presently, the main welding method on the market is direct welding by heating platform. Direct welding means that the heating platform is preheated to the welding temperature, and then the substrate is put in the fixture on the heating platform, so that the aluminum substrate is closely linked to the heating platform. Due to the good thermal conductivity of aluminum substrate, the direct welding method can make solder paste reach the melting point quickly and shorten the time needed for welding. However, using this method, the solder joint is exposed to air, and the solder paste is easily oxidized during the welding process. Vacuum furnace welding improves upon that method. During vacuum furnace welding, the filament is placed in the vacuum furnace, the vacuum is activated and then, according to the heating curve, the furnace is slowly set to the temperature [7–14]. Relatively speaking, vacuum furnace welding avoids the oxidation of the solder. Although much research has been conducted about different welding methods used on LED, there is limited literature reporting relevant studies for the comparison of direct welding and vacuum furnace welding. In order to select the most suitable welding method according to actual production, the effect of two welding methods on the properties of FC-LED filaments was studied in this paper [15–17].

In this study, shearing force testing was conducted using the strength tester. The microstructure of the fracture surface was observed by SEM. The steady-state voltage and the luminous flux of filament were measured by integration sphere, respectively. Then, the two filaments were lit and aged with a constant current source of 285 V/15 mA. The steady-state voltage, the luminous flux and the main wavelength were measured after aging for 72, 216, 336, and 504 h, respectively. By using the above testing methods, the shearing force, the microstructure of the solder joint after the shear failure test, the steady-state voltage, the luminous flux, and the main wavelength curve were obtained, respectively. By summarizing and analyzing the above data, the effects of two different welding methods on the performance of the FC-LED filament were obtained. Subsequently, the suitable welding methods can be selected according to the different production requirements.

## 2. Sample Preparation and Testing

### 2.1. Sample Preparation

Enraytek EA0820A (Enraytek, Shanghai, China) FC-LED chips with Au layer were soldered to the substrate. The size of the chip was 8 mil × 20 mil, the working voltage was 3.1–3.2 V, and the wavelength was 452.5–455.0 nm (Figure 1a). For the study, aluminum substrate was used with a length of 255 mm, and the number of the chips that could be placed was 204. After packaging, the rated voltage of the filament was 260 V. Moreover, Sn-3.0Ag-0.5Cu (SAC305) solder (Earlysun, Shenzhen, China) was used; the main components were Sn/Ag/Cu and a fluxing agent, and the melting point of the solder was 217–220 °C. To avoid influencing the experiment results because of different solder sizes, the solder was uniformly printed on each pad of the substrate using the HTGD-GTS (HTGD, Suzhou, China) type solder paste printer. Then, with the help of a JUST BT-929-type (JUST, Xiamen, China) LED bonder, the chip was placed onto the substrate printed with SAC305 solder until it was finished. The heating platform was heated to 270 °C in advance, then half of the filaments were put directly in the fixture on the heating platform for 60 s. The other filaments were placed in the TORCH RS220+ (TORCH, Beijing, China) type vacuum welding furnace, and after vacuum, the temperature was raised to 270 °C according to the temperature curve (Figure 1b). The welding time was 25 min.

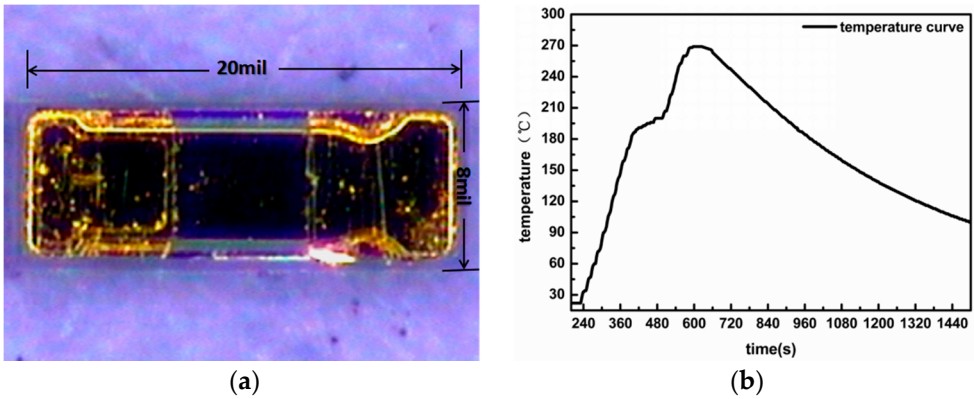

<div align="center">(<b>a</b>)　　　　　　　　　　　　　　　　　(<b>b</b>)</div>

**Figure 1.** (**a**) Flip-chip light-emitting diode (FC-LED) chip physical diagram, (**b**) temperature curve.

## 2.2. Sample Testing

One filament was randomly selected from the two groups of samples, respectively. The shearing force test was conducted using a XYZTEC strength tester (XYZTEC, Shenzheng, China). The microstructure of the fracture surface was observed by scanning electron microscope (SEM) (Hitachi, Tokyo, Japan). Ten filaments were randomly selected from each of the two groups and lit with a constant current source of 285 V/15 mA, and the filament was defined as steady-state after 15 min of continuous lighting. The voltage and the luminous flux of the filament were measured by an LED300E (EVERFINE, Hangzhou, China) integration sphere, respectively. Then, the two filaments were lit and aged with a constant current source of 285 V/15 mA, respectively. Aging data were measured at 72, 216, 336, and 504 h.

## 3. Results and Discussion

### 3.1. Shearing Force Scatter Diagram and Microstructure of Fracture Surface

Figure 2 shows the chip shearing force of direct welding (DW) and vacuum furnace welding (VFW). It can be seen that the shearing force of the chip using the two welding methods is higher than the standard shearing force of 200 gf; all of the shearing force met the requirements of the chip shearing force for FC-LED filaments. According to the average shearing force of the chip, the DW group was 306.08 gf and the VFW group was 356.46 gf. The VFW group was 16.5% higher than the DW group [18,19].

Figure 3 shows the microstructure of the solder joint after the shear failure test. As can be seen from Figure 3, the microstructure of the solder joint using direct welding is very rough, and the cracks and voids are distributed; the solder joint using vacuum furnace welding was smooth. Combined with the shearing force of the two chip groups in Figure 2, it can be seen that the cracks and voids in the solder joint have a negative effect on the shearing force.

In the case of the chip welded using the vacuum furnace, and because the temperature follows curve heating during the welding process, the flux in the solder reaches the volatile temperature and then volatilizes, after which the solder reaches the melting point and then melts. Compared with the instant melting of the solder using direct welding, vacuum furnace welding is more favorable for fluxing agent volatilization, and it can effectively avoid solder joint porosity caused by fluxing agent volatilization over time. Additionally, the welding environment of the vacuum furnace is a vacuum, which effectively prevents the oxidation of the elements in the solder at high temperature and avoids the production of cracks at the solder joint. Therefore, the shearing force of the VFW group is higher than that of the DW group [20–22].

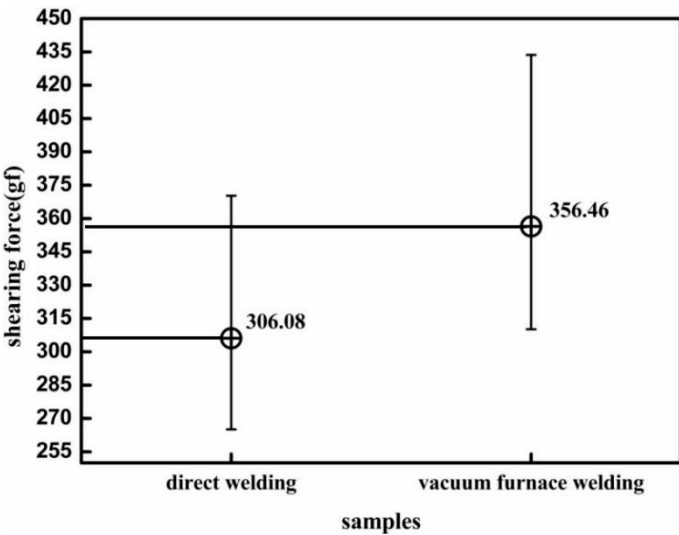

**Figure 2.** Average shear strength of 10 FC-LED filaments in two sample package.

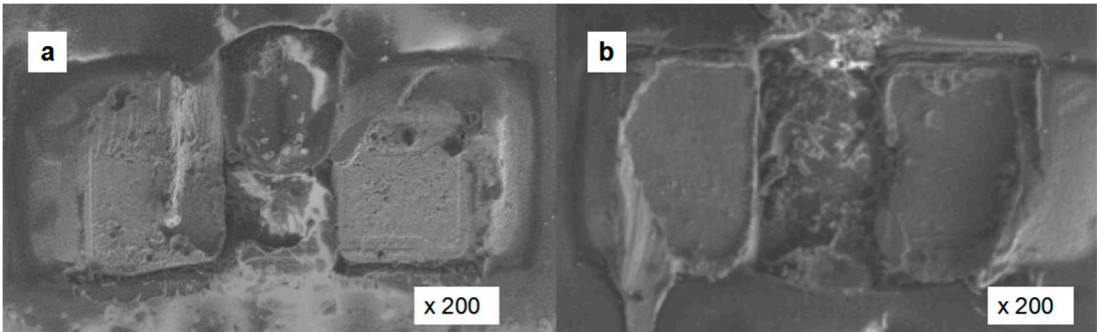

**Figure 3.** Microstructure of the solder joint after the shear failure test using (**a**) direct welding and (**b**) vacuum furnace welding.

*3.2. Steady-State Voltage and Blue Light Luminous Flux*

Figure 4 shows the box plot of the steady-state voltage of 10 FC-LED filaments after lighting 15 min. It can be seen from the diagram that most of the first group have higher voltages than the entire latter group. The average voltage of the DW group was 261.44 V, and the average voltage of the VFW group was 260.72 V. *D*-value was less than 1%. However, according to the formula of variance:

$$\sigma^2 = \frac{\sum(X - \mu^2)}{N},$$

$$(1)$$

where $\sigma^2$ is the variance, which can reflect the degree of dispersion of the data; X is the value of each sample; $\mu$ is the total average value; $N$ is the sample number. The authors calculated the variance of the previous set of voltages $\sigma^2_1 = 0.884$, the latter $\sigma^2_2 = 0.0225$. $\sigma^2_1 \gg \sigma^2_2$ illustrates that the steady-state voltage of the direct welding filament is more discrete, whereas the vacuum furnace welding steady-state voltage is more concentrated. It can be concluded that there is a big difference in voltage between directly welded filaments. During the practical use of the filament, if it is used in parallel or in series, it may accelerate the aging of the filament because of its partial pressure or shunt unevenness, and shorten the service life of the filament.

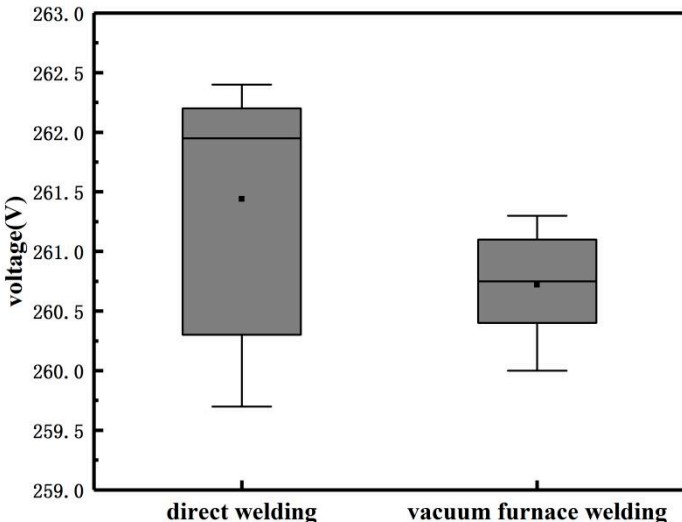

**Figure 4.** Steady-state voltage measured for 10 FC-LED filaments after 15-min lighting.

Figure 5 is the steady-state luminous flux of two groups of filaments. It can be seen from the figure that the luminous flux of the VFW group is larger than that of the DW group. The average luminous flux in the DW group was 94.8 lm, and the average luminous flux in the VFW group was 101.0 lm, higher by 6.2 lm than the previous group. In the packaging process of white light LED, the conversion ratio of blue light luminous flux to white light is approximately 1:7. Therefore, assuming that suitable phosphors are chosen, the integral luminous flux of filaments using vacuum furnace welding is much higher than that using direct welding.

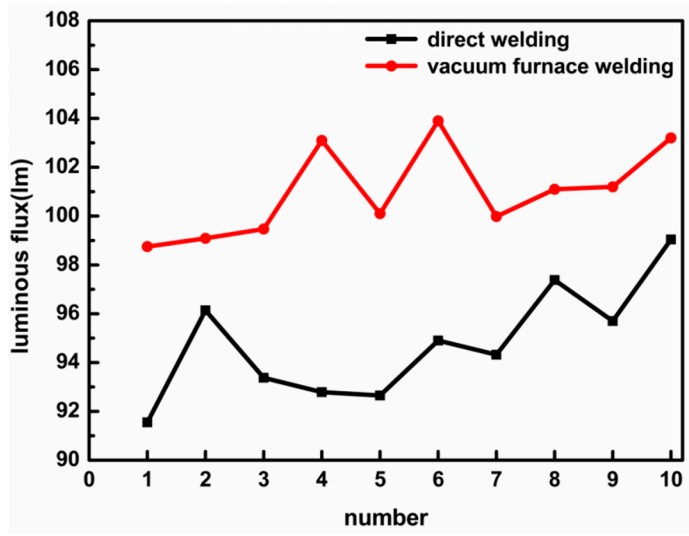

**Figure 5.** Steady-state luminous flux measured for 10 FC-LED filaments after 15-min lighting.

This occurs because the soldering temperature rises too quickly and the insufficient volatilization of the fluxing agent in the solder causes voids in the solder layer; the solder joint is exposed to the air and oxidizes to form an aperture gap, so that the resistance of the chip increases and the voltage increases. In addition, the size and number of the voids in each filament vary from one to the other, which causes the resistance of each filament to be different, further reflected in that fact that the voltage dispersion degree becomes larger. By comparison, the solder fluxing agent in the vacuum furnace is volatilized relatively completely, the solder joint is not oxidized, the voids formed in the solder layer are less porous, the resistance of the filament is relatively consistent, and the voltage fluctuation is also

small. At the same time, because the defects of the DW group are large and the internal resistance increases, the current density through the luminous region decreases when the filament is lit under the same conditions, leading to the low luminous flux [23,24].

### 3.3. Change of Photoelectric Performance with Aging Time

Figure 6 shows the steady-state voltage curve of the filaments with aging time increasing. From this figure, it can be seen that the voltage of the two groups of filaments decreases significantly after 72 h of aging. This is because in the early stage of the filaments' aging process, a large amount of heat is produced in the solder layer, which causes a chemical reaction in the solder to form an intermetallic compound. The solder becomes more compact, the hole becomes smaller, the resistance value of the filament is reduced, and the current is constant when it is lit. The decrease of resistance leads to the decrease of voltage.

The voltage then increases slowly during the aging of 72–504 h, but the overall trend is still downward. Before and after aging, the absolute value of the steady-state voltage change rate of the DW group was 1.03%, and the steady-state voltage change rate of the VFW group was 1.07%. There is a small difference between the two groups.

In the process of falling luminous flux, there is sometimes an abnormal rise. It was concluded that there were two contradictory mechanisms in the aging process of LED, that is, the increase mechanism of optical output and the mechanism of attenuation. The authors of [25] believe this is because in the early stages of aging, the P-type doped layer MG-H complex matrix decomposition caused by the activation of Mg and the conductivity of the P-type layer is increased, and the forward voltage drops; on the other hand, the composite probability of minority carriers increases, thereby enhancing the light output. In the meantime, with the increase of the impurity level and the non-radiative recombination of the center, much new energy is generated in the process of aging, resulting in a decreased light output, thus increasing luminous flux and the attenuation of two kinds of phenomena [26].

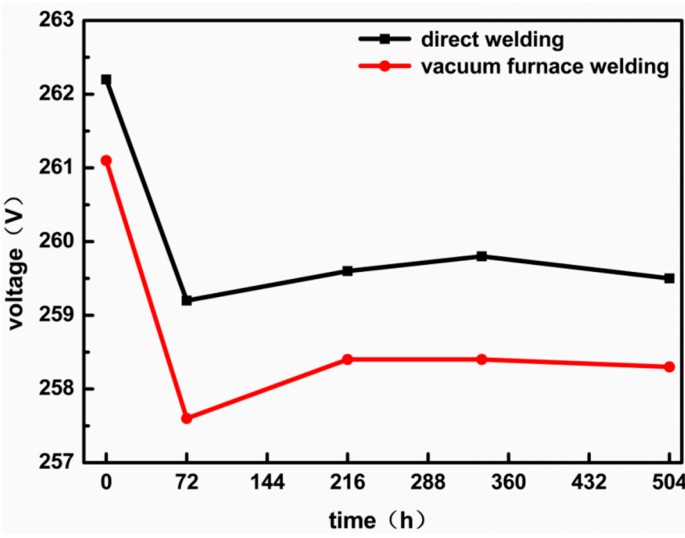

**Figure 6.** Steady-state voltage curve of the filaments with aging time increasing.

Figure 7 shows the change curve of aging time and steady-state luminous flux. As was the case with the voltage, the luminous flux of the two groups decreased obviously at 72 h. After aging for 216 h, the luminous flux of the DW group increased continuously, whereas that of the VFW group increased first and then decreased. The steady-state luminous flux of both groups decreased after aging. The luminous flux decreased by 8.45% in the DW group and 5.4% in the VFW group. In fact, 50% of the initial luminous flux for the LEDs was used as indicators and 70% for general lighting applications; in this study, the 70% value was selected in light of the type of applications for which these LEDs were

used. They all met the standard. However, compared with the direct welding filament, the vacuum furnace welding filament had a higher luminous flux maintenance. Therefore, the light decay after vacuum welding is less.

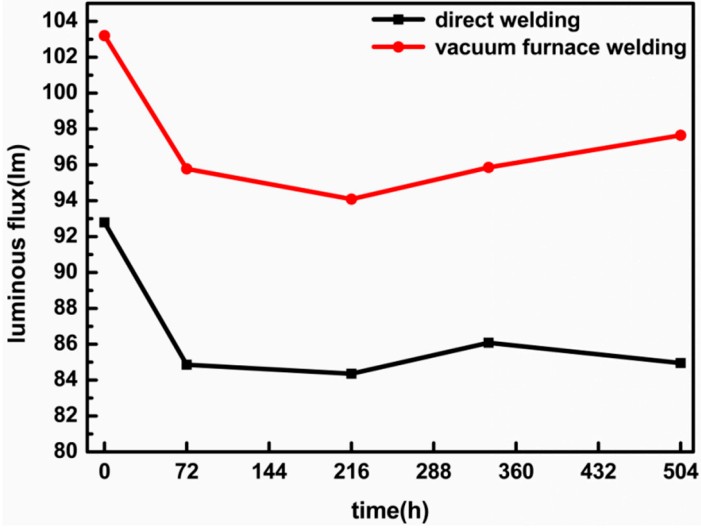

**Figure 7.** Change curve of aging time and steady-state luminous flux.

## 4. Conclusions

After direct welding and vacuum furnace welding of the FC-LED filaments, the shearing force was tested. After the shearing force test, the microstructure of the fracture surface, the steady-state voltage, and the luminous flux were measured, respectively. Then, the filaments were lit and aged with a constant current source of 285 V/15 mA, respectively. Aging data were measured at 72, 216, 336, and 504 h. By comparing and analyzing the two groups of data, it was concluded that:

(a) the shearing force of the two groups followed the standard, but the average shearing force of VFW group was higher than that of the DW group and the microstructure of VFW group fault was smoother, and the pores and voids were fewer and smaller;

(b) in terms of steady-state voltage and luminous flux, the VFW group had a more concentrated voltage and a higher luminous flux;

(c) the steady-state voltage of both groups decreased first and then increased slowly, but the voltage change rate was not very different from that before and after aging;

(d) both luminous flux maintenance rates were within the standard range, but the VFW group rates were higher.

In conclusion, if there is a higher requirement for filament in a practical application, such as the filament is connected in series or in parallel and needs a higher luminous flux, it can be welded using vacuum furnace welding. If the focus in on production efficiency and the high performance of filaments is not required, direct welding can be used.

**Author Contributions:** Conceptualization, J.Z.; Methodology, J.Z.; Software, M.L.; Validation, M.L., W.L. and B.Y.; Formal Analysis, M.L.; Investigation, M.L.; Resources, J.Z.; Data Curation, M.L.; Writing-Original Draft Preparation, M.L.; Writing-Review & Editing, M.L.; Visualization, M.S.; Supervision, W.W. and B.G.; Project Administration, B.Y.; Funding Acquisition, J.Z.

**Funding:** This research was funded by [the Science and Technology Planning Project of Zhejiang Province, China] grant number [2018C01046], [Enterprise-funded Latitudinal Research Projects] grant number [J2016-141, J2017-171, J2017-293, J2017-243], [Sponsored by Shanghai Sailing Program] grant number [18YF1422500], and [Research start-up project of Shanghai Institute of Technology] grant number [YJ2018-9].

**Acknowledgments:** This work was supported by the Science and Technology Planning Project of Zhejiang Province, China (2018C01046), Enterprise-funded Latitudinal Research Projects (J2016-141), (J2017-171), (J2017-293), (J2017-243); sponsored by the Shanghai Sailing Program (18YF1422500); a research start-up project of the Shanghai Institute of Technology (YJ2018-9).

**Conflicts of Interest:** The authors declare no conflicts of interest.

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
