# Peer review of "Effect of Different Welding Methods on Flip-Chip LED (FC-LED) Filament Properties"

_applsci, doi:10.3390/app8112254_

Round 1
Reviewer 1 Report
In this work, the authors compared 2 different methods for FC-LED filament welding. Overall, the manuscript was well organized and the supporting data/results were clearly presented. My biggest concern regarding publishing this work is the English writing, there were certain grammar mistakes and incorrect wording, which could really confuse the readers. My comments are as follows:
1) Line 70: "The microstructure of the fracture surface is destroyed by SEM." What do you mean by "destroyed"? should that be "inspected"?
2) There is no "、" in English, use "," when separating numbers (e.g. Line 74: 72, 216, 336 and 504 h) and use "." when referring to certain figures (e.g. Line 86: Figure 1.a);
3) In Figure 3, the two images should be at the same magnification;
4) Need to add more descriptions of the figure to the figure captions. E.g. Figure 4, instead of "steady-state voltage", say "Steady-state voltage measured for 10 FC-LED filaments after 15-min lighting". Same thing for Figure 2 and 5;
5) Since the authors were calculating the variance for 10 different filaments using DW and VFW, I would recommend to use box plot instead of line plot in figure 4;
6) Line 181, "the resistance of the filament is relatively concentrated", should "concentrated" be "consistent"? Check the wording
Again, overall writing really needs to be improved, accept after addressing comments and improving writing.
Author Response
Response to Reviewer 1 Comments
Point 1: Line 70: "The microstructure of the fracture surface is destroyed by SEM." What do you mean by "destroyed"? should that be "inspected"?
Response 1: I have replaced "destroyed" of the two sentences in the article with "observed". (Line 70, 105)
Point 2: There is no "、" in English, use "," when separating numbers (e.g. Line 74: 72, 216, 336 and 504 h) and use "." when referring to certain figures (e.g. Line 86: Figure 1.a);
Response 2: I have modified the use of the five punctuation errors in the article. (Line 74, 86, 98, 111 and 235)
Point 3: In Figure 3, the two images should be at the same magnification;
Response 3: I've changed it in the manuscript. (Line 139)
Point 4: Need to add more descriptions of the figure to the figure captions. E.g. Figure 4, instead of "steady-state voltage", say "Steady-state voltage measured for 10 FC-LED filaments after 15-min lighting". Same thing for Figure 2 and 5;
Response 4: I have modified and add more descriptions of the figure to the figure captions. (Figure 4, 2 and 5)
Point 5: Since the authors were calculating the variance for 10 different filaments using DW and VFW, I would recommend to use box plot instead of line plot in figure 4;
Response 5: I use box plot instead of line plot in figure 4 and changed the corresponding descriptive sentences in the manuscript. (Line 145-147)
Point 6: Line 181, "the resistance of the filament is relatively concentrated", should "concentrated" be "consistent"? Check the wording.
Response 6: After checking, I have corrected "concentrated" to "consistent". (Line 183)

Reviewer 2 Report
This manuscript “ Effect of Different Welding Methods on Flip-chip LED (FC-LED) Filaments Properties” is the description and develop of TSV technology.
This is a good technical work. The main problem it is the writing with defects in wording. The text needs a general review of wording.
All the sections have relevant information, and the order it is correct, but need improve the wording, especially in the Conclusions that are confused.
These are my comments:
a) Line 83, In the first phrase “Enraytek EA0820A FC LED chips with Au diodes were soldered”, the words “Au diodes” are correct?.
b) Line 93, In the phrase “JUST BT-929-type solid crystal machine” this is a pick and . place LED machine. It is the JUST “BT-929 type,”?. Explain the words “solid crystal machine”, I see it out of context.
c) Line 94, SAC it is SAC305.
d) Line120, 356.46gf VFW group is 16.5% higher than DW group. This phrase need a comma or words, for best comprehension.
e) The wording of the paragraph should be improved from line 127 to 136. Then a vacuum are repeat very near. Similar wording problem between lines 150 to 158.
f) Figure 3 need horizontal auxiliary lines for identify the shearing force.
g) Line 153. σ21>σ2, really it is >>, more of ten times greater.
h) Line 208. The Fire 7 and phrase “The same as voltage, the luminous flux of two groups decreased obviously at 72h. 208 After aging for 216 hours, the luminous flux of DW group increased continuously, 209 while that of VFW group increased first and then decreased. “ are not consistent. Review the explanation.
i) The wording of the conclusion should be improved. If you use bullets of word for every test conclusion, all will be clear. Other solution are separate the phrase of each test conclusion.
j) In figure use. First letter on capital letter. You can observe other papers for copy these characteristics and facilitate the work editor’s.
In the reference the word [J]1313
Author Response
Response to Reviewer 2 Comments
Point 1: Line 83, In the first phrase “Enraytek EA0820A FC LED chips with Au diodes were soldered”, the words “Au diodes” are correct?.
Response 1: I changed the “Au diodes” in the sentence to “Au layer”. (Line 84)
Point 2: Line 93, In the phrase “JUST BT-929-type solid crystal machine” this is a pick and place LED machine. It is the JUST “BT-929 type,”?. Explain the words “solid crystal machine”, I see it out of context.
Response 2: The type of machine mentioned in the sentence is “JUST BT-929-type” and change “solid crystal machine” to more specialized terminology “LED bonder”. LED bonder is a machine that can place the chip on the substrate through solder. (Line 93)
Point 3: Line 94, SAC it is SAC305.
Response 3: I've changed it in the manuscript. (Line 94)
Point 4: Line120, 356.46gf VFW group is 16.5% higher than DW group. This phrase need a comma or words, for best comprehension.
Response 4: I have changed the sentence in the manuscript: DW group is 306.08gf and VFW group is 356.46gf, VFW group is 16.5% higher than DW group. (Line 119-120)
Point 5: The wording of the paragraph should be improved from line 127 to 136. Then a vacuum are repeat very near. Similar wording problem between lines 150 to 158.
Response 5: I've changed it in the manuscript.
Point 6: Figure 3 need horizontal auxiliary lines for identify the shearing force.
Response 6: I added horizontal auxiliary lines for identify the shearing force. (Line 137)
Point 7: Line 153. σ21>σ2, really it is >>, more of ten times greater.
Response 7: I've changed it in the manuscript. (Line 154)
Point 8: Line 208. The Fire 7 and phrase “The same as voltage, the luminous flux of two groups decreased obviously at 72h. 208 After aging for 216 hours, the luminous flux of DW group increased continuously, 209 while that of VFW group increased first and then decreased. “ are not consistent. Review the explanation.
Response 8: “The same as voltage” just mean when the aging time is 0-72 hours, aging data of voltage and luminous flux are all reduced.
Point 9: The wording of the conclusion should be improved. If you use bullets of word for every test conclusion, all will be clear. Other solution are separate the phrase of each test conclusion.
Response 9: I divided the conclusion into four parts, a-d, as requested. (Line 237-247)
Point 10: In figure use. First letter on capital letter. You can observe other papers for copy these characteristics and facilitate the work editor’s.
Response 10: I've changed it in the manuscript.

Reviewer 3 Report
The manuscript describes about importance of two welding methods which are useful on Flip-Chip LED filaments. For current and next generation, this kind of work is very imoprtant. In this report two welding methods studied carefully and explained use of each method as per requirements. This report is recommended to publish in applied sciences after the following minor revisions.
1. In page no 2, line 40, there should be reference where you mentioned energy-saving, environmental protection, high reliability, flexible design and other advantages, has been wide developed and applied in the field of lighting
2. Please polish the text from line 47-49 and 86-89, and make sure it has good flow
3. Please provide the better SEM image of fig 3a. This image has lot of electron flow (White color). Please retake this image and get better one. If you cant improve it as such, I suggest you to do sputtering (with gold) and retake SEM image.
4. There are few words repeated. For example, in line 146, group group and in line 157, partial partial. Please fix these minor errors
5. In line 188, From the figure should be changes as From this figure
6. I found at most of the places, weird notation for commas. For example in line 74, aging 72、216、336、504h. Please fix these in entire manuscript
7. In Figure 3, please assign the what are the figure a and figure b
8. In figure 6, it mention that the voltage of the two groups of 188 filaments decreases significantly after 72 hours of aging and then increases slowly 189 during aging of 72h-504 h. Please provide the explaination for voltage decrease.

Author Response
Response to Reviewer 3 Comments
Point 1: In page no 2, line 40, there should be reference where you mentioned energy-saving, environmental protection, high reliability, flexible design and other advantages, has been wide developed and applied in the field of lighting
Response 1: I added reference[4] to this place. (Line 40)
Point 2: Please polish the text from line 47-49 and 86-89, and make sure it has good flow
Response 2: I've changed it in the manuscript.
Point 3: Please provide the better SEM image of fig 3a. This image has lot of electron flow (White color). Please retake this image and get better one. If you cant improve it as such, I suggest you to do sputtering (with gold) and retake SEM image.
Response 3: I have re-added figure 3a that meets the requirements.
Point 4: There are few words repeated. For example, in line 146, group group and in line 157, partial partial. Please fix these minor errors
Response 4: I've changed it in the manuscript.
Point 5: In line 188, From the figure should be changes as From this figure
Response 5: I've changed it in the manuscript. (Line 190)
Point 6: I found at most of the places, weird notation for commas. For example in line 74, aging 72、216、336、504h. Please fix these in entire manuscript
Response 6: I have modified the use of the five punctuation errors in the article. (Line 74, 86, 98, 111 and 235)
Point 7: In Figure 3, please assign the what are the figure a and figure b
Response 7: I have assigned figure a and b in Figure 3.
Point 8: In figure 6, it mention that the voltage of the two groups of 188 filaments decreases significantly after 72 hours of aging and then increases slowly 189 during aging of 72h-504 h. Please provide the explaination for voltage decrease.
Response 8: This is because in the early stage of the filaments aging process, a large number of heat is produced in the solder layer, which causes chemical reaction between the solder to form intermetallic compound. The solder becomes more compact, the hole becomes smaller, the resistance value of the filament is reduced and the current is constant when it is lighten. The decrease of resistance leads to the decrease of voltage. (Line 191-196)
